# Impact of Clinical Decision Support System Assisted prevention and management for Delirium on guideline adherence and cognitive load among Intensive Care Unit nurses (CDSSD-ICU): Protocol of a multicentre, cluster randomized trial

**Shan Zhang**[1]☯, **Shu Ding**[1,2]☯, **Wei Cui**[1], **Xiangyu Li**[1], **Jun Wei**[3], **Ying Wu**[1] *

1 School of Nursing, Capital Medical University, Beijing, China, 2 Cardiology Department, Beijing Chao-Yang Hospital, Capital Medical University, Beijing, China, 3 Respiratory Intensive Care Unit, Xuanwu Hospital, Capital Medical University, Beijing, China

☯ These authors contributed equally to this work.
* helennywu@vip.163.com

**Data Availability Statement:** No datasets were generated or analysed during the current study. All

## Abstract

### Background

Adherence to the delirium bundle intervention is sub-optimal in routine practice, and inappropriate use of the instructional design of interventions may result in higher cognitive load among nurses. It remains unclear whether the Clinical Decision Support System (CDSS) Assisted Prevention and Management for Delirium (*CDSS-AntiDelirium*) results in the improvement of adherence to delirium intervention and the reduction of extraneous cognitive load, as well as improving adherence to delirium intervention, among nurses in the intensive care unit (ICU).

### Methods

This study (named the CDSSD-ICU) is a multicentre, prospective, cluster randomized controlled clinical trial. A total of six ICUs in two hospitals will be randomized in a 1:1 ratio to receive either the *CDSS-AntiDelirium* group or the delirium guidelines group. The *CDSS-AntiDelirium* consists of four modules: delirium assessment tools, risk factor assessment, a nursing care plan, and a nursing checklist module. Each day, nurses will assess ICU patients with the assistance of the *CDSS-AntiDelirium*. A total of 78 ICU nurses are needed to ensure statistical power. Outcome assessments will be conducted by investigators who are blinded to group assignments. The primary endpoint will be adherence to delirium intervention, the secondary endpoint will be nurses' cognitive load measured using an instrument to assess different types of cognitive load. Repeated measures analysis of variance will be used to detect group differences. A structural equation model will be used to clarify the mechanism of improvement in adherence.

relevant data from this study will be made available upon study completion.

**Funding:** This research was supported by Grant 72304196 from the National Natural Science Foundation of China. The funders had and will not have a role in study design, data collection and analysis, decision to publish, or preparation of the manuscript.

**Competing interests:** The authors have declared that no competing interests exist.

## Discussion

Although the CDSS has been widely used in hospitals for disease assessment, management, and recording, the applications thereof in the area of delirium are still in infancy. This study could provide scientific evidence regarding the impact of a CDSS on nurses' adherence and cognitive load and promote its further development in future studies.

## Clinical trial registration

ChiCTR1900023711 (Chinese Clinical Trial Registry).

## Introduction

Delirium, a common complication among patients in the intensive care unit (ICU), with incidence ranging from 70 to 87% [1, 2], is associated with a longer length of hospital stay and increased mortality [3, 4]. Therefore, the Pain, Agitation, Delirium, Immobility, and Sleep (PADIS) guidelines recommend the use of bundle interventions as a way to prevent and manage ICU delirium, which focuses on eliminating delirium risk factors [5]. However, adherence to the delirium bundle intervention is sub-optimal in routine clinical care [6]. Previous studies have demonstrated that various barriers may hinder adherence to implementing the bundle intervention, such as heavy workload and burden of maintaining nursing care records [7], complexity algorithm of assessment tools [8], as well as numerous risk factors that must be collected through multiple channels [9]. These factors may contribute to lowering the speed for receiving and processing information, decreased capacity of working memory, and a greater cognitive load [10, 11].

Cognitive load refers to the total amount of cognitive resources that a person needs to process cognitive activities [12] and comprises three load types: intrinsic, extraneous, and germane cognitive [13, 14]. Intrinsic cognitive load refers to the demand for working memory resources in processing information elements and their interactions in related cognitive activities [15]. Extraneous cognitive load is produced by inappropriate task presentation other than the cognitive activity itself [16]. Germane cognitive load promotes schema construction, thereby facilitating the execution of cognitive activities [15]. Higher extraneous cognitive load has been identified as one of the most important problems in providing intensive care [17, 18]. This negatively impacts nurses, such as by reducing their capacity to acquire knowledge [18, 19].

Considering the negative consequences of a higher cognitive load owing to the implementation of delirium bundle intervention, it is important to provide ICU nurses with a tool that can reduce extraneous cognitive load. Clinical decision support systems (CDSSs) have been widely used in most hospitals worldwide for disease assessment, management, and recording [20, 21]. Several studies have shown that a CDSS can help medical staff by reducing the amount of clinical information that must be recalled, which leads to a significant reduction in cognitive load [22, 23]. Therefore, we developed the **C**linical **D**ecision **S**upport **S**ystem (CDSS) **A**ssisted Pre-ve**nti**on and Management for **Delirium** (*CDSS-AntiDelirium*) based on the Cognitive Load Theory and Human Factors Engineering (HFE) [24], which were used to guide the division of functional modules, process design, and user interface (UI) design throughout the development process. There were five steps to designing and developing CDSS-AntiDelirium, including (1) assessing the reasons for non-adherence to the PADIS guidelines and the needs of users regarding functionality, workflow, and UI of the CDSS-AntiDelirium; (2) designing the

functional modules following PADIS guidelines and nursing processes; (3) iteratively designing the system structure and UI of the CDSS-AntiDelirium; (4) improving the CDSS-AntiDelirium based on end-user feedback in pilot testing; and (5) usability testing of the CDSS-AntiDelirium. Development of the CDSS-AntiDelirium was completed by a multidisciplinary team, including healthcare professionals, nursing researchers, nursing informaticians, and information technology engineers. The CDSS-AntiDelirium was tested and showed good usability. However, the impact of CDSS-AntiDelirium on adherence and cognitive load in ICU nurses is still unclear. Therefore, in this study, we aim to conduct a multicentre, cluster randomized trial (RCT) to investigate the effectiveness of the CDSS-AntiDelirium on improving adherence to delirium intervention and reducing cognitive load among ICU nurses as well as reveal the mechanism of adherence improvement in delivering delirium intervention.

## Materials and methods

The protocol is developed according to the Standard Protocol Items: Recommendations for Interventional Trials (SPIRIT), see S1 File [25].

### Study design

The study is a multicentre, cluster RCT. A total of six ICUs in two hospitals will be randomized in a 1:1 ratio to receive either a CDSS-AntiDelirium group or an individual intervention group. Fig 1 shows the design of the study.

### Study setting

Six ICUs in two tertiary hospitals are selected based on a high incidence of delirium as well as variety in the samples, including two surgical ICUs (SICUs), two respiratory ICUs (RICUs), and two cardiac ICUs (CICUs). Hospital A is a 1000-bed university-affiliated teaching hospital in Beijing, with 52 ICU beds in three participating ICUs. Hospital B has 730 beds and three eligible ICUs with 46 beds, in which the annual admission rate is around 850 patients. These two hospitals have both fully implemented hospital informatization.

### Participants

Registered nurses eligible for the study (1) have a minimum of 1 year of experience working in the ICU; (2) work full-time in the unit; and (3) consent to participate in this study. Nurses will be excluded from the study if they are on leave for various reasons during the study period.

### Recruitment

All recruitment will be carried out by trained research staff who will not be involved in the intervention and will be blinded to the nurses' group assignments. To identify eligible nurses, researchers will administer a questionnaire among all nurses to screen nurses based on the inclusion and exclusion criteria. Written informed consent will be obtained from all eligible nurses. To retain participating nurses, research staff will explain the benefits for nurses if this study is successfully implemented.

### Randomization and blinding

The risk of between-group contamination is reduced with cluster randomization. Therefore, ICUs will be randomized 1:1 to receive either CDSS-AntiDelirium care or PADIS guideline care. The allocation sequence is based on computer-generated random numbers and allocation

| | STUDY PERIOD | | | | |
| --- | --- | --- | --- | --- | --- |
| | Enrolment | Allocation | Post-allocation | | Close-out |
| **TIMEPOINT** | $-t_1$ | **0** | $t_1$ *(baseline)* | $t_2$ *(post-intervention)* | $t_3$ *(1 month)* |
| **ENROLMENT:** | | | | | |
| **Eligibility screen** | X | | | | |
| **Informed consent** | X | | | | |
| **Allocation** | | X | | | |
| **INTERVENTIONS:** | | | | | |
| *AI-AntiDelirium* | | | ●━━━━━● | | |
| *Paper-based PADIS guideline* | | | ●━━━━━● | | |
| **ASSESSMENTS:** | | | | | |
| *Demographics questionnaire* | X | X | | | |
| *Adherence to ICU delirium assessment* | | | | X | X |
| *Adherence to delirium risk factors assessment* | | | | X | X |
| *Adherence to delirium prevention or management intervention* | | | | X | X |
| *Cognitive load* | | | | X | X |

**Fig 1. Schedule of enrolment, interventions, and assessments.** *AI-AntiDelirium*: the **A**rtificial **I**ntelligence **A**ssisted Preve**nt**ion and Management for **Delirium** *PADIS guideline*: the Pain, Agitation, Delirium, Immobility, and Sleep (PADIS) Guidelines.

is performed by a statistician who is independent of the data analyses and not involved in data collection. All eligible nurses will be recruited until the sample size is sufficiently large.

Investigators who will enroll participants will not be made aware of the randomization list. Once the ICUs (cluster) that will take part in the study have been determined, randomization will be performed, considering the size of each ICU and the type of disease treated. The following criterion will be considered in this process: whether the center is classified as having a high incidence of delirium. Investigators will recruit participants following predefined inclusion and exclusion criteria. Baseline data and endpoint measures will be obtained by data collectors or outcome assessors who have no role in the intervention and are blinded to the allocations.

However, this will not be possible for blind nurses who implement the interventions to the allocation owing to the nature of the intervention.

### Intervention group

The appropriate subset of interventions from the PADIS guideline (such as the ABCDEF bundle) will be tailored to each patient's specific risk factors using the *CDSS-AntiDelirium*. The intervention is targeted to eliminate various modifiable risk factors of ICU delirium, including hearing impairment (e.g., speaking loudly, slowly, and patiently with the patient; assisting the patient to correctly wear hearing aids), visual impairment (e.g., assisting the patient to wear eyeglasses correctly), pain (e.g., providing non-pharmacological interventions, such as distraction and relaxation therapy), use of anesthetic or sedatives (e.g., implementation of the spontaneous awakening trial), mechanical ventilation (e.g., conducting spontaneous breathing trials according to the doctor's orders), indwelling catheter (e.g., removing the catheter as soon as possible), infection (e.g., reducing invasive operations and avoiding unnecessary catheterization), immobility (e.g., helping patients to do passive range-of-motion exercises, 10 times for each joint), sleep deprivation (e.g., assisting patients to wear earplugs or anti-noise equipment), and no visits from family members (e.g., encourage family visits). A set of IF-THEN rules are stored in the CDSS-AntiDelirium, which are designed to trigger the nursing care plan based on the identified delirium risk factors. These interventions are delivered 7 days a week primarily by nurses in a face-to-face mode, until the patient's discharge. Nursing care activities will vary among specific frequency, duration, intensity, or dose.

Nurses in the intervention group will provide delirium assessment and intervention with the assistance of the CDSS-AntiDelirium (Fig 2). The CDSS-AntiDelirium consists of four modules: delirium assessment tools, risk factor assessment, nursing care plan, and a nursing checklist module. Before the study, an educational program is to be delivered by researchers (Fig 1), including knowledge regarding ICU delirium risk factors, assessment tools such as the Confusion Assessment Method for the Intensive Care Unit (CAM-ICU), and bundle interventions. Additionally, eligible ICU nurses will be trained in how to operate the CDSS-AntiDelirium. The CDSS-AntiDelirium will be installed on a personal digital assistant, for which nurses must complete the necessary registration information including hospital, department, name, and password. Nurses can then log into the system and (1) enter the *Delirium assessment tools* module to perform ICU delirium assessment at least two times (8:00–10:00 a.m., 4:00–6:00 p.m.) per day; (2) enter the *Risk factor assessment* module to assess patient's existing risk factors (e.g., hearing impairment, visual impairment, pain, and mechanical ventilation), in which dynamic delirium prediction rule for ICU patients (DYNAMIC-ICU) will be utilized to classify patients into different risk levels [26]; (3) enter the *Nursing care plan* module to confirm the plan, which is automatically tailored based on the results of risk factor assessment; (4) enter the *Nursing checklist* module to check the specific execution timetable of each intervention based on the results of the nursing care plan. The data transformation algorithms are predetermined.

Each day, the nurse will assess ICU patients with the assistance of the CDSS-AntiDelirium (Table 1).

### Control group

Nurses in the control group will provide delirium intervention based on a paper-based delirium assessment tool (CAM-ICU), risk factor assessment, and bundle interventions. Before the study, an educational program is to be delivered by researchers (Fig 2); the content is the same

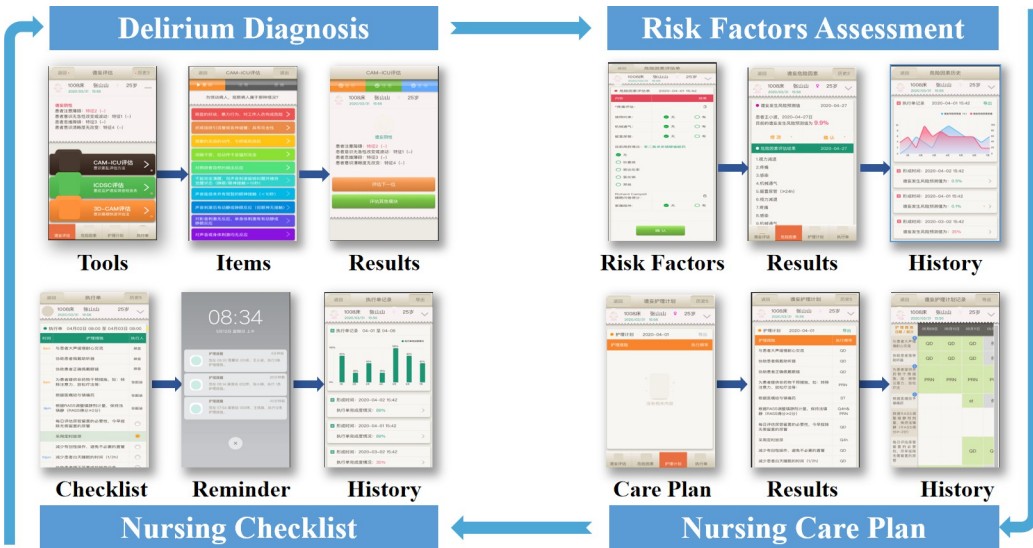

**Fig 2. User interface of four modules of the CDSS-AntiDelirium.**

as that in the intervention group. In addition, nurses are trained to use paper-based individual interventions. During the study, the intervention will be provided by nurses (see Table 1).

## Outcome measures

Outcome assessment will be implemented by trained research staff who are not involved in the clinical nursing care of patients.

The primary outcome is adherence to delirium intervention among ICU nurses, which is defined as adherence to implementing delirium and risk factor assessment, delirium prevention, and management intervention. The reasons for non-adherence will be recorded daily by the intervention staff. A complete adherence rate is recorded as 100%, and a complete non-adherence rate is recorded as 0%. The adherence rate is between 0 and 100, according to the following formulae.

**Table 1. Safety and quality pathway for ICU nurses in implementing delirium intervention.**

| Step | Actions | Intervention group | Control group |
|------|---------|--------------------|---------------|
| Step 1 | Delirium diagnosis | The nurse completes the item one by one according to the prompts in *CDSS-AntiDelirium*, and the system will automatically present whether the patient has ICU delirium or not (See S2 File). | The nurse will use the paper-based delirium assessment tools to identify delirium (See S3 File). |
| Step 2 | Identify delirium risk factors | The nurse fills out a brief risk factor assessment in *CDSS-AntiDelirium*, including three categories, predisposing, disease-related, iatrogenic, and environmental factors; the system will automatically report personalized risk factors of each patient and show the risk prediction value based on a dynamic ICU delirium prediction rule (See S2 File). | The nurse will complete a paper-based risk factors evaluation list to identify the risk factors of each patient, including three categories same as to intervention group (See S3 File). In addition, the nurse will calculate the delirium risk prediction value based on the delirium prediction rule. |
| Step 3 | Tailor nursing care plan | The nurse will receive the personalized delirium prevention or management care plan automatically which is tailored based on the results of the risk factors assessment (See S2 File). | The nurse needs to tailor an appropriate subset of interventions from the PADIS guidelines that target patients' specific risk factors (See S3 File). |
| Step 4 | Implement nursing checklist | The nurse can check the nursing checklist, which automatically displays the specific execution time of each intervention based on the results of the nursing care plan (See S2 File). | The nurse needs to make sure of the execution time of each intervention from the ICU delirium prevention and treatment interventions sheet (See S3 File). |

a. *Adherence to delirium interventions* is calculated as (the number of interventions implemented/the total number of interventions expected) × 100%. The total number of interventions that should be implemented is in accordance with the nursing care plan. For example, an individualized delirium care activity checklist with 10 interventions is tailored by the CDSS-AntiDelirium based on the results of the risk factor assessment. However, a bedside nurse only completes six interventions during the whole shift; therefore, the adherence to delirium intervention is (6/10) × 100% = 60%. During a whole shift, a bedside nurse will take care of three to five ICU patients; therefore, the total adherence to delirium intervention per day is calculated as (number of interventions implemented for all patients cared for/total number of interventions expected for all patients cared for) × 100%.

b. *Adherence to risk factor assessment* is defined as (the number of risk factors assessed/the total number of risk factor assessments expected) × 100%. The total number of risk factors that should be assessed in accordance with the risk factor assessment sheet, including hearing impairment, visual impairment, pain, use of anesthetic or sedatives, mechanical ventilation, indwelling catheter, infection, immobility, sleep disorders, and no visits from family members. The total adherence to risk factor assessment per day is calculated as (the number of risk factors assessed for all patients cared for/the total number of risk factor assessments expected for all patients cared for) × 100%.

c. *Adherence to ICU delirium assessment* is defined as (the number of ICU delirium assessments/the total number of ICU delirium assessments expected) × 100%. The total number of ICU delirium assessments for each patient is calculated two times (8 to 10 A.M. and 4 to 6 P.M.) per day. The total adherence to ICU delirium assessment per day is calculated as (number of ICU delirium assessments for all patients cared for/total number of ICU delirium assessments expected for all patients cared for) × 100%.

The secondary outcomes include overall and three types of cognitive load for nurses. Cognitive load is to be assessed using an instrument for measuring different types of cognitive load (MDT-CL) with 10 items [27]. Each item is scored from 0 to 10; the higher the score, the greater the cognitive load. The overall cognitive load is equal to the average score of the three cognitive loads. The Chinese version retains sensitivity and specificity in measuring cognitive load [28]. Cronbach's α of the Chinese version of the MDT-CL is 0.818; Cronbach's α value in the measurement of intrinsic cognitive load (Items 1, 2, and 3) is 0.879; Cronbach's α value in the measurement of extraneous cognitive load (Items 4, 5, and 6) is 0.878; and Cronbach's α in the measurement of germane cognitive load (Items 7, 8, 9, and 10) is 0.946.

Level of knowledge refers to nurses' knowledge of ICU delirium assessment tools, risk factors, prevention, and management intervention and is measured using the questionnaire on ICU delirium knowledge. A 20-item single-choice questionnaire has been developed [29], with Cronbach's α for the questionnaire of 0.82. The higher the score, the higher the knowledge level regarding ICU delirium.

## Participant timeline

Enrollment and data collection started in November 2022. Recruitment will continue until the target population (78 nurses) is enrolled, which is expected to end in June 2023. Afterward, the data analysis will be carried out for subsequent publication.

## Data collection

Before the study, uniform training will be delivered by research staff to data collectors, including theoretical knowledge and assessment skills for delirium, to maintain high consistency of

delirium assessment in all included units. To minimize error and maximize reliability, the project director will perform the following: providing intensive training to the assessors, including a review of the procedure outlined based on a standardized patient to ensure high inter-rater reliability (kappa > 0.9); meeting with the assessors every week to review procedures and check the quality of assessment for the primary outcome. Data collectors will recruit nurses based on the inclusion and exclusion criteria, and informed consent will be obtained from all eligible nurses before allocation (Fig 1). Then, basic demographic data of ICU nurses will be collected, and baseline cognitive load will also be assessed.

During the study, adherence to delirium intervention and the cognitive load of nurses will be measured at the end of each shift on each day (Fig 1).

## Data management

All data will be collected on a printed, pre-coded form that will be double entered into an electronic database and go through extensive checking for errors and data completeness. The ICUs involved in the study will only have access to their own data. We will perform the following to ensure data quality. (1) Before the study, all investigators and data collectors will be trained to master the procedures of data collection. (2) All participating nurses will be trained to apply the tools, but different groups of nurses will attend a different education session: intervention group nurses will learn how to use the CDSS-AntiDelirium, and control group nurses will learn how to apply paper-based individual interventions. (3) Data cleaning and reviewing will be ongoing to detect missing data and inconsistencies. The investigator will solve any problems promptly.

## Data analysis

PASS version 11.0.7 (NCSS, LLC. Kaysville, Utah, USA) will be used to estimate the sample size for the current study. This cluster RCT aims to improve adherence to delirium intervention among ICU nurses using the CDSS-AntiDelirium as compared with PADIS guidelines. We assume that the adherence is 80% in the CDSS-AntiDelirium group and 50% in the paper-based individual intervention group, based on a similar study that explored adherence to delirium intervention [30]. Sample size calculations show that 12 nurses in each cluster will provide 80% power and a two-sided significance level ($\alpha$) of 0.05, with an intra-cluster (within-unit) correlation of 0.002 [31]. In considering a possible attrition rate of 10%, we will enroll 13 nurses in each ICU setting; therefore, the final sample size for the study is 78 (13 × 6) ICU nurses.

All data will be analyzed using IBM SPSS version 21.0 (IBM Corp., Armonk, NY, USA) with an intention-to-treat (ITT) principle. To avoid the reduction in statistical efficiency and bias caused by missing data, multiple imputations will be performed to create a complete dataset, assuming that data are lost at random. Statistical analyses will be done by a statistician who is blinded to the intervention allocations. Continuous variables (e.g., age, years of ICU experience) will be described as mean and standard deviation for normally distributed data and as median and interquartile range for non-normally distributed data. Comparisons between groups will be performed with the Mann–Whitney U-test (analysis of variance) or Wilcoxon test, including cognitive load, and adherence. Categorical variables (e.g., sex, education level) will be expressed as frequency and percentage. The chi-square test or Fisher's exact test will be used to examine between-group differences according to demographics.

Considering the correlation with repeated measures in the same individual, repeated measures analysis of variance (RMANOVA) will be used to analyze data to assess the effect of time and group on adherence and different types of cognitive load that are collected every day

during the study period. Mauchly's W statistic will be used to assess the estimated sphericity and Greenhouse–Geisser correction will be used if the data do not conform to the sphericity test ($P < 0.05$). The influence of potential covariates, e.g., age, sex, educational level, marital status, department, years of ICU experience, and level of professional title, will be assessed by including their interaction with the intervention in RMANOVA. Bonferroni post hoc tests will be used to assess within-group changes over time, as well as between-group differences during the study. In addition, adherence and cognitive load will also be analyzed using a linear mixed model for repeated measures (MMRM); the combination of RMANOVA and MMRM will allow us to draw more reliable and valid conclusions. Subgroup analyses for the primary end-point (adherence to delirium interventions) will be performed in the following groups: age ($< 30$ years vs. $\geq 30$ years), years of ICU experience ($< 10$ years vs. $\geq 10$ years), education level, professional title, baseline intrinsic cognitive load, extraneous intrinsic cognitive load, and germane cognitive load (median value as the cut-off point).

AMOS 21.0 statistical software for path analysis will be used to analyze the relationship among task presentation (different groups), cognitive load, and adherence. The acceptable model goodness of fit is as follows: (1) $\chi^2$/degrees of freedom (df) ratio ($\chi^2$/df) $< 3$; (2) goodness-of-fit index (GFI) $> 0.85$; (3) comparative fit index (CFI) $> 0.90$; (4) Tucker–Lewis Index (TLI) $> 0.90$; and (5) root mean square error of approximation $< 0.05$ [32]. Effect sizes are measured using the standardized coefficients and are evaluated at the following levels: 0.1–0.3 = small; 0.3–0.5 = medium; $> 0.5$ = large [33]. A bootstrap approach with 95% bias-corrected confidence intervals (CIs) and 95% Percentile CIs (based on 2,000 bootstrap samples) will be used to test the significance of the mediation effect. All tests are two-tailed, and a $P$-value $< 0.05$ is considered statistically significant.

## Data monitoring

Supervisors will adopt a safety role to monitor the quality and completeness of data in each center. They will audit the original data and clarify any problems with the data in the collection (for example, insufficient enrolment or retention of participants, inadequate researchers, and missing data). The supervisor has the right to terminate the trial for reasons of patient safety. There is no interim analysis in this trial, and the study will continue until the target sample size is reached.

## Harms

There is no harm expected to nurses participating in this study. In the present study, the patient interventions will not pose an additional risk to patients. However, adverse events can occur even during normal care that can result in patient dysfunction, signs of discomfort, prolonged hospitalization, as well as life-threatening events. Any adverse events (e.g., fall, pressure ulcer, unplanned extubation, bradycardia, tachycardia) will be recorded by data collectors and reported to the Ethics Committee of the hospital as soon as possible.

## Patient and public involvement

Patients and/or the public (including clinical professionals) will be involved in the study design, or conduct, or reporting of this research. A participatory workshops carried out in October 2022 focused on collecting their opinions and solving problems in the design of the protocol. Information about the study (CDSSD-ICU) is used to inform patients and public representatives. The findings of the CDSSD-ICU will be summarized for publication.

## Ethical considerations and dissemination

The Ethics Committee of the two participating hospitals approved the study protocol (#XW082 and #CY1225-33). Approval from the ethics committee must be obtained for any modification to this protocol. The trial was registered in the Chinese Clinical Trial Registry (www.chictr.org.cn) with the identifier ChiCTR1900023711 on June 8, 2019. During the study, guidelines of Good Clinical Practice will be fully followed to guarantee the rights of participants.

   All study data are anonymized and treated confidentially and will not be disclosed to anyone other than members of the research group. For the duration of the study, all paper documents for each participant will be stored in locked cabinets, and electronic data will be entered into a password-protected database. The results of the study will be submitted for publication in peer-reviewed journals. All data underlying the findings will be shared through the Baidu network (https://pan.baidu.com/s/1qVeHCHE23n75eGWJBOz1ZQ). All authors will take part in reviewing drafts of the manuscript and solving any other publication problems in this study. In addition, we propose disseminating the results of this study at nursing or delirium-related academic conferences; all academic posters and papers will be written by research team members.

## Ancillary and post-trial care

Because the ICUs involved in this study will incorporate the intervention protocol into routine care, no additional compensation is received by ICU nurses.

## Validity and reliability/Rigour

A cluster RCT with practical maneuverability will be conducted to implement the trial, and high-quality evidence will be provided from the results of this study. This study adopts a rigorous methodology to minimize the risk of potential bias. For instance, before the study, all data collectors blinded to the group allocation will be trained to use the assessment tools to proficiently and effectively evaluate the variables to minimize errors and improve reliability. The operational definitions of variables involved in this study will be predefined by reviewing the literature and guidance to minimize information bias. The outcome assessors will receive intensive training in terms of validated and reliable instruments to assess adherence, cognitive load, and other outcomes, which can minimize observer bias.

## Results and discussion

During the implementation of delirium intervention, ICU nurses require greater cognitive resources to maintain constant attention, recall information, and conduct real-time monitoring [22]. ICU nurses also must conduct complex decision-making, implying a high cognitive load, which may result in poor adherence to the clinical guidelines. However, a CDSS can automatically collect, sort, and classify patient information and provide immediate feedback to healthcare providers during disease diagnosis, treatment, and nursing activities [34]. Therefore, ICU nurses can enhance their adherence to delirium intervention with the assistance of a CDSS. However, no complete CDSS exists to assess, prevent, and intervene in ICU delirium, and the impact of a CDSS on nurses' outcomes remains unclear. The CDSS-AntiDelirium was designed to identify delirium and modifiable risk factors in patients and to automatically provide personalized delirium preventive and management interventions. Use of the CDSS-AntiDelirium may encourage ICU nurses to be more involved in implementing delirium interventions.

To the best of our knowledge, this would be the first multicentre, cluster randomized trial focused on improving understanding of whether the CDSS-AntiDelirium is effective and beneficial for adherence to delirium intervention and reducing cognitive load among ICU nurses, as well as revealing the mechanism of adherence improvement in delivering delirium intervention. We hypothesize that: (1) the CDSS-AntiDelirium can reduce the extraneous cognitive load of ICU nurses and promote adherence to implementing delirium interventions, and (2) extraneous cognitive load is a mediating variable between task presentation and adherence to delirium intervention. The results of this study will have a profound impact on the research field regarding Cognitive Load Theory and CDSSs and will promote further development in future studies.

Several limitations will exist in the proposed study. First, this study will only be carried out in two hospitals in a single region, which will limit the generalization of our results. We should conduct further research in more hospitals in more regions. Second, the potential for the risk of contamination involves nurses from different groups in the same hospital who may communicate with each other during their non-working hours. Therefore, before the study, we will carry out a training program and ask nurses in the two groups to avoid discussing this study during breaks. Finally, long-term outcomes are not designed to be assessed in our study. Despite challenges, these limitations will be reduced using rigorous methodology. The issues in this study should be addressed in future studies to further improve efficiency and effectiveness.

## Conclusions

The findings from this study will highlight the effectiveness of the CDSS-AntiDelirium for adherence to delirium intervention and cognitive load among ICU nurses and will encourage hospital administrators to adopt a CDSS for complex activities. In addition, this study will clarify the mechanism of the CDSS-AntiDelirium in improving adherence to delirium intervention among ICU nurses and will provide a theoretical and methodological basis for improving adherence to guidelines.

## Supporting information

**S1 File. SPIRIT 2013 checklist.**
(PDF)

**S2 File. Clinical Decision Support System Assisted prevention and management for Delirium (*CDSS-AntiDelirium*).**
(PDF)

**S3 File. Paper-based individual interventions.**
(PDF)

**S4 File. Research project submitted to the ethics committee (English).**
(PDF)

**S5 File. Research project submitted to the ethics committee (Chinese).**
(PDF)

**S6 File.**
(PDF)

## Acknowledgments

We extend our sincere thanks to Professor Meihua Ji (School of Nursing, Capital Medical University in Beijing, China) for her editing assistance.

## Author Contributions

**Conceptualization:** Shan Zhang, Shu Ding, Wei Cui, Xiangyu Li, Ying Wu.

**Data curation:** Xiangyu Li, Jun Wei.

**Investigation:** Shan Zhang, Wei Cui, Xiangyu Li, Jun Wei.

**Methodology:** Shan Zhang, Shu Ding, Ying Wu.

**Project administration:** Shan Zhang.

**Resources:** Shan Zhang, Shu Ding.

**Supervision:** Shan Zhang, Jun Wei, Ying Wu.

**Writing – original draft:** Shan Zhang, Shu Ding, Wei Cui, Xiangyu Li, Jun Wei, Ying Wu.

**Writing – review & editing:** Ying Wu.

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
