## [Decision Letter · Decision Letter 0]

25 Jun 2023

PONE-D-23-02468Impact of Artificial Intelligence Assisted Prevention and Management for Delirium on cognitive load in Intensive  Care Unit nurses (AID-ICU): protocol of a multicenter, cluster randomized trialPLOS ONE

Dear Dr. Wu,

Thank you for submitting your manuscript to PLOS ONE. After careful consideration, we feel that it has merit but does not fully meet PLOS ONE’s publication criteria as it currently stands. Therefore, we invite you to submit a revised version of the manuscript that addresses the points raised during the review process.

 The protocol has been assessed by two peer reviewers and both point out important aspects to be addressed in a revision. Importantly, these include the rationale of the study, the reporting of outcome measures and the rationale and the nature of the intervention. Please make sure that these aspects are clearly addressed and clarified.

We look forward to receiving your revised manuscript.

Kind regards,

Sascha Köpke

Academic Editor

PLOS ONE

Journal Requirements:

NO authors have competing interests

Reviewers' comments: **Comments to the Author**

1. Does the manuscript provide a valid rationale for the proposed study, with clearly identified and justified research questions?

Reviewer #1: No

Reviewer #2: Yes

2. Is the protocol technically sound and planned in a manner that will lead to a meaningful outcome and allow testing the stated hypotheses?

Reviewer #1: No

Reviewer #2: Partly

3. Is the methodology feasible and described in sufficient detail to allow the work to be replicable?

Reviewer #1: No

Reviewer #2: No

4. Have the authors described where all data underlying the findings will be made available when the study is complete?

Reviewer #1: No

Reviewer #2: No

5. Is the manuscript presented in an intelligible fashion and written in standard English?

Reviewer #1: No

Reviewer #2: Yes

6. Review Comments to the Author

Reviewer #1: Dear authors,

thank you very much for the opportunity to review the submitted manuscript. The manuscript includes a protocol of a RCT assessing the impact of an artificial intelligence delirium-tool on cognitive load of critical care nurses.

I have reviewed hundreds of scientific manuscripts, but I have never read a manuscript that insults and belittles nurses in intensive care units in such a way. The very first sentence, "Higher cognitive load of nurses is a common problem in the intensive care units, which is associated with nurses’ poor performance of activities, as well as related to poor patient outcomes," implies that nurses are often cognitively overloaded, unable to do their jobs cognitively, and therefore harm patients. It is an insult to all nurses who work hard, take justifiable pride in their work, and study and gain further qualifications on the side. For ethical and professional policy reasons, this manuscript cannot be approved for publication.

The authors' reasoning is inconsequential. The rationale cites literature in which physicians as well as nurses have been studied in cognitive load in simulation centres, with no differences. Other literature cites consensus plans and curricula. There is no evidence of cognitive overload in nurses.

Further, primary and secondary outcome parameters are confounded for this study, which is actually a QI project.

Reviewer #2: Thank you very much for the very interesting protocol. Please bear in mind that my suggestions are meant to improve the manuscript. I hope you feel your work is appreciated.

You would have to clarify why the outcome cognitive load is described in the title of the manuscript, although it is described as a secondary outcome in the text.

Intervention AI-AntiDelirium:

• Z.85: You write that the AI AntiDelirium was developed on the basis of Cognitive Load Theory. Please describe this process.

• Please briefly describe the basic operation of AI. If necessary, a conceptual differentiation from clinical decision support systems (CDSS) would be necessary.

• Are data transformation algorithms predetermined or adaptive?

• Who was involved in the development of AI?

• Some information is still missing to describe the intervention. You could orientate yourself on the TiDier checklist. In particular, it would be interesting to know whether the intervention takes place in the form of an APP on a tablet or possibly integrated into a patient data management system.

• The algorithm linking the identified delirium risk factors to the Nursing Care Plan should be described.

Dynamic ICU Delirium Prediction Rule

• At what APACHE-II score are 3 points assigned? At a cut-off point above 7.5 (Fan et al. 2019)?

• The weighting of a very high score for Hearing Deficits (9 points) and Indwelling Catheter (8 points) can be derived from the preliminary study. How empirically reliable is this information in general and in particular with very wide confidence intervals?

Control Group

• You mention the PADIS guideline, but the description of the individual interventions or bundle interventions would be useful.

Adherence:

• Perhaps it would be useful to describe the calculation of adherence (a-c), with an example and/or a formula.

• Z.177: Is actual delirium risk factors assessment meant here?

• Z.172: Is this about adherence to the delirium risk factors assessment in general, or to the assessment of individual risk factors?

• Z.225: You assume an adherence of 80% for the AI-AntiDelirium group and 50% for the paper-based PADIS group and refer to the study by Trogrlic et al. 2018. This assumption should be explained.

---

## [Author Response · Author response to Decision Letter 0]

11 Jul 2023

Reviewers' comments: Comments to the Author

1. Does the manuscript provide a valid rationale for the proposed study, with clearly identified and justified research questions?

Reviewer #1: No

Reviewer #2: Yes

Response: We modified the relevant sentence in the introduction to make it more clear. As following:

However, the impact of AI-AntiDelirium on adherence and cognitive load in ICU nurses is still unclear. Therefore, this study aims to conduct a multicenter, cluster randomized trial (RCT) to investigate the effectiveness of AI-AntiDelirium on improving adherence to delirium intervention and reducing cognitive load of ICU nurses and reveal the mechanism of adherence improvement in delivering the delirium intervention.

2. Is the protocol technically sound and planned in a manner that will lead to a meaningful outcome and allow testing the stated hypotheses?

Reviewer #1: No

Reviewer #2: Partly

Response: We added the relevant sentence in the Materials and methods. As following:

Before the study, uniform training will be delivered by research staff to data collectors, including theoretical knowledge and assessment skills for delirium, in order to maintain high consistency of delirium assessment in all included units. To minimize error and maximize reliability, the project director performed the following: providing intensive training to the assessors including a review of the procedure outlined based on a standardized patient to ensure high inter-rater reliability (Kappa > 0.9); meeting with the assessors every week to review procedures and check the quality of the assessment for the primary outcome.

In order to avoid the reduction in statistical efficiency and bias caused by missing data, multiple imputations was performed to create a complete dataset assuming that data is lost at random.

The influence of potential covariates, e.g., age, gender, educational level, marital status, departments, years of ICU experience, and level of professional title, will be collected by including their interaction with the intervention in RMANOVA. 

3. Is the methodology feasible and described in sufficient detail to allow the work to be replicable?

Reviewer #1: No

Reviewer #2: No

Response: We provided more detailed information in the revised manuscript to allow the work to be replicable based on the Template for Intervention Description and Replication (TIDieR) checklist, please see the detailed information in the “Intervention group and control group, Table 1, Figure 2, S3 File, S4 File and sample size calculations”.

4. Have the authors described where all data underlying the findings will be made available when the study is complete?

Reviewer #1: No

Reviewer #2: No

Response: All data underlying the findings will be shared through the Baidu network (https://pan.baidu.com/s/1qVeHCHE23n75eGWJBOz1ZQ). We added this sentence in the “dissemination” section.

5. Is the manuscript presented in an intelligible fashion and written in standard English?

Reviewer #1: No

Reviewer #2: Yes

Response: 

To present the manuscript in an intelligible fashion and written in standard English, Professor Meihua Ji (United States Registered Nurse) guided us in English writing and had good writing skills. She graduated from the University of Pittsburgh School of Nursing in 2018 and worked in the School of Nursing, Capital Medical University in 2019 through the Overseas Talent Introduction Program. We extend our thanks to Professor Meihua Ji in the Acknowledgments section. As following:

We extend our sincere thanks to Professor Meihua Ji ( School of Nursing, Capital Medical University in Beijing, China) for her editing assistance.

6. Review Comments to the Author

Reviewer #1: Dear authors,

thank you very much for the opportunity to review the submitted manuscript. The manuscript includes a protocol of a RCT assessing the impact of an artificial intelligence delirium-tool on cognitive load of critical care nurses.

I have reviewed hundreds of scientific manuscripts, but I have never read a manuscript that insults and belittles nurses in intensive care units in such a way. The very first sentence, "Higher cognitive load of nurses is a common problem in the intensive care units, which is associated with nurses’ poor performance of activities, as well as related to poor patient outcomes," implies that nurses are often cognitively overloaded, unable to do their jobs cognitively, and therefore harm patients. It is an insult to all nurses who work hard, take justifiable pride in their work, and study and gain further qualifications on the side. For ethical and professional policy reasons, this manuscript cannot be approved for publication.

The authors' reasoning is inconsequential. The rationale cites literature in which physicians as well as nurses have been studied in cognitive load in simulation centres, with no differences. Other literature cites consensus plans and curricula. There is no evidence of cognitive overload in nurses.

Further, primary and secondary outcome parameters are confounded for this study, which is actually a QI project.

Response: Thank you very much for pointing out this key issue. 

Firstly, we are very sorry for the misunderstanding caused by our expression, as we all know, nursing care in the intensive care units is characterized by elevated demand on workloads and increased cognitive load. ICUs require higher levels of nurse staffing than other health care environments, because of the demands of close monitoring and observation of patients and necessity of managing technological life support equipment for severely ill patients, which can ensure the quality of care and patient safety. We have great respect for ICU nurses, and we are part of the team of nurses. We modified the relevant sentence in the abstract and introduction. As following: 

[Abstract]

Adherence to the delirium bundle intervention is sub-optimal in routine practice, and inappropriate use of the instructional design of interventions may result in higher cognitive load among nurses.

[Introduction]

Higher extraneous cognitive load has been identified as one of the most important problems in providing intensive care. It negatively impacts nurses, such as reduced learning capacity for acquiring knowledge.

Secondly, as for the cites literature about the evidence of cognitive overload in nurses, we substituted more suitable literature. As following:

Higher extraneous cognitive load has been identified as one of the most important problems in providing intensive care[17,18]. It negatively impacts nurses, such as reduced learning capacity for acquiring knowledge[17,19].

Reference

[17] B. Han, D. B. Gal, M. Mafla, L. D. Sacks, A. T. Singh, and A. Y. Shin. 2021. Role of Texting as a Source of Cognitive Burden in a Pediatric Cardiovascular ICU. Hosp Pediatr (2021), e253-e257. DOI 10.1542/hpeds.2021-005869

[18] Tovar LO Gomez, A. M. Henao-Castano, and I. Y. Troche-Gutierrez. 2021. Prevention and treatment of delirium in intensive care: Hermeneutics of experiences of the nursing team. Enferm Intensiva (Engl Ed) (2021). DOI 10.1016/j.enfi.2021.05.001

[19] C. Yu, J. Jiang, M. Zhong, H. Zhang, and X. Duan. 2023. Training load of newly recruited nurses in Grade-A Tertiary Hospitals in Shanghai, China: a qualitative study. BMC Nurs. (2023), 9. DOI 10.1186/s12912-022-01138-z

Thirdly, we are very sorry that our expressions in the title and the methodology were not clear, which caused reviewer to be confused about the primary outcome and the secondary outcome. We have made the corrections in the Title. As following:

 “Impact of Artificial Intelligence Assisted Prevention and Management for Delirium on guideline adherence and cognitive load in Intensive Care Unit nurses (AID-ICU): protocol of a multicenter, cluster randomized trial”.

Reviewer #2: Thank you very much for the very interesting protocol. Please bear in mind that my suggestions are meant to improve the manuscript. I hope you feel your work is appreciated.

You would have to clarify why the outcome cognitive load is described in the title of the manuscript, although it is described as a secondary outcome in the text.

Response: Thanks a lot for your very helpful comment. In our study, the primary outcome will be adherence to delirium guideline intervention, the secondary endpoint will be the cognitive load. We revised the title into Impact of Artificial Intelligence Assisted Prevention and Management for Delirium on guideline adherence and cognitive load in Intensive Care Unit nurses (AID-ICU): protocol of a multicenter, cluster randomized trial. 

Intervention AI-AntiDelirium:

• Z.85: You write that the AI AntiDelirium was developed on the basis of Cognitive Load Theory. Please describe this process.

Response: We added more information about the development of AI-AntiDelirium in the introduction section, as following:

Therefore, we developed the Artificial Intelligence Assisted Prevention and Management for Delirium (AI-AntiDelirium) based on the Cognitive Load Theory and Human factors engineering (HFE), which were used to guide the division of functional modules, process design, and user interface design throughout the development process. There are five steps to design and develop AI-AntiDelirium, including (1) assessing the reasons for non-adherence to the PADIS guidelines and the needs of users for the functionality, workflow, and UI of AI-AntiDelirium; (2) designing the functional modules following PADIS guidelines and nursing processes; (3) iteratively designing the system structure and the UI of AI-AntiDelirium; (4) improving the AI-AntiDelirium based on end-users feedback in a pilot testing; (5) usability testing of AI-AntiDelirium. 

• Please briefly describe the basic operation of AI. If necessary, a conceptual differentiation from clinical decision support systems (CDSS) would be necessary.

Response: We added the information about the basic operation of the AI-AntiDelirium in the “Intervention group” section, and also in Figure 1 and Table 1, as following:

AI-AntiDelirium is installed on Personal Digital Assistant (PDA), where nurses need to complete necessary registration information including hospital, department, name, password, etc. And then nurses can log into the system and (1) Enter the delirium assessment tools module to perform ICU delirium assessment at least two times (8:00-10:00 a.m., 4:00-6:00p.m.) per day; (2) Enter the risk factor assessment module to assess patient's existing risk factors, and a dynamic delirium prediction rule for ICU patients (DYNAMIC-ICU) would be utilized to classify patients into different risk levels; (3) Enter the nursing care plan module to confirm the plan, which is automatically tailored based on the results of risk factors assessment; (4) Enter the nursing checklist module to check the specific execution timetable of each intervention based on the results of the nursing care plan. 

• Are data transformation algorithms predetermined or adaptive?

Response: The data transformation algorithms are predetermined.

• Who was involved in the development of AI?

Response: The design and development of the AI-AntiDelirium were completed by a multidisciplinary team, which included healthcare professionals (clinical experts, ICU nurses) as end-users, nursing researchers, nursing informaticians, and IT engineers (UI designers, architects, programmers) who specialized in developing clinical nursing information systems and the health-related mobile applications. The team members continued to discuss online daily and meet weekly to discuss any revisions required throughout the whole process. As for the detailed information about the development of the AI-AntiDelirium was described in another article, therefore, we provide the information in this response letter.

• Some information is still missing to describe the intervention. You could orientate yourself on the TiDier checklist. In particular, it would be interesting to know whether the intervention takes place in the form of an APP on a tablet or possibly integrated into a patient data management system.

Response: Thanks a lot for your professional and helpful comment. We added more information based on the Template for Intervention Description and Replication (TIDieR) checklist. As following:

The appropriate subset of interventions from the PADIS guideline (such as the ABCDEF bundle) will be tailored to the patient's specific risk factors by AI-AntiDelirium. The intervention is targeted at eliminating various modifiable risk factors of ICU delirium, including hearing impairment (speaking loudly, slowly, and patiently with the patient; assisting the patient to wear hearing aids correctly, etc.), visual impairment (assisting the patient to wear eyeglasses correctly, etc.), pain (provide non-pharmacological interventions, such as distraction and relaxation therapy, etc.), use of anesthetic or sedatives (implementation of the spontaneous awakening trial, etc.), mechanical ventilation (conduct spontaneous breathing trials according to the doctor's order, etc.), indwelling catheter (remove the catheter as soon as possible, etc.), infection (reduce invasive operations and avoid unnecessary catheterization, etc.), immobility (help patients to do passive range-of-motion exercises, 10 times for each joint, etc.), sleep deprivation (assist patients to wear earplugs or anti-noise equipment, etc.), and no family members visit (family visits are encouraged, etc.). These interventions are delivered 7 days a week primarily by nurses in a face-to-face mode, until the patient’s discharge. Nursing care activities varied with a specific frequency, duration, intensity, or dose. 

AI-AntiDelirium is installed on Personal Digital Assistant (PDA), where nurses need to complete necessary registration information including hospital, department, name, password, etc. 

• The algorithm linking the identified delirium risk factors to the Nursing Care Plan should be described.

Response: Thank you very much for the comment. We described the algorithm linking the identified delirium risk factors to the nursing care plan, as following:

A set of IF-THEN rules in AI-AntiDelirium were stored and designed to trigger the nursing care plan based on the identified delirium risk factors.

Dynamic ICU Delirium Prediction Rule

• At what APACHE-II score are 3 points assigned? At a cut-off point above 7.5 (Fan et al. 2019)?

Response: We revised the sentence to make it easier to understand in Dynamic ICU Delirium Prediction Rule section in the S3 File, as following:

Predictors included in the model including the APACHE-II score above 7.5, history of chronic disease, sleep deprivation, use of anesthetic or sedatives, infection, indwelling catheter, and hearing impairment. The simplified score for each predictor was 3, 3, 3, 3, 4, 8, and 9 points. 

• The weighting of a very high score for Hearing Deficits (9 points) and Indwelling Catheter (8 points) can be derived from the preliminary study. How empirically reliable is this information in general and in particular with very wide confidence intervals?

Response: In our previous article[1], a dynamic delirium prediction rule in

patients admitted to the Intensive Care Units (DYNAMIC-ICU) were constructed.

DYNAMIC-ICU has the highest predictive value of stratifying ICU patients into different levels of risk in delirium development with the C statistic of 0.907 based on seven risk factors and 0.888 based on total risk scores of seven factors in the derivation cohort (n=336) and 0.900 in the validation cohort (n=224). In order to allow us to draw more reliable and valid conclusions, firstly, all patients were undergone extensive assessment with interviews and a review of the medical record to collect necessary information on possible risk factors. Secondly, given that our study was an observational cohort study, other confounding factors would bias the model results. To control bias as much as possible, we conducted univariate analysis and logistic regression, and the final model performed well in both the derivation and external validation cohorts. Thirdly, we applied mean imputation and bootstrap resampling procedures to augment data for deriving predictors in our prediction rule. These advanced statistical methods may minimize the effects of missing data and limit model overfitting to ensure more stable variables are included for prediction rule establishment and validation. In the future, we can further examine the role of different risk factors in the dynamic delirium prediction rule.

[1]Fan H，Ji M，Huang J，et al. Development and validation of a dynamic delirium prediction rule in patients admitted to the Intensive Care Units (DYNAMIC-ICU): A prospective cohort study. Int J Nurs Stud, 2019, 93: 64-73.

Control Group

• You mention the PADIS guideline, but the description of the individual interventions or bundle interventions would be useful.

Response: Thanks very much for your professional suggestion. We revised the sentence to make it clear. As following:

“In addition, nurses are trained to use paper-based individual interventions.”

And the description of the control group (paper-based individual interventions) is uniformly modified in the revised manuscript.

Adherence:

• Perhaps it would be useful to describe the calculation of adherence (a-c), with an example and/or a formula.

Response: Thank you very much for your kind suggestion. We described the calculation of adherence (a-c) with a formula. As following:

a.Adherence to delirium interventions is calculated as (the number of interventions implemented / the total number of interventions expected) × 100%. The total number of interventions that should be implemented in accordance with the nursing care plan. For example, an individualized delirium care activity checklist with 10 interventions is tailored by AI-AntiDelirium based on the results of the risk factors assessment. However, the bedside nurse only completes 6 interventions during the whole shift, therefore, the adherence to delirium interventions is (6/10) × 100% = 60%. During a whole shift, a bedside nurse will take care of three to five ICU patients, therefore, the total adherence to delirium interventions per day is calculated as (the number of interventions implemented for all cared patients/ the total number of interventions expected for all cared patients) × 100%.

b.Adherence to risk factors assessment is defined as (the number of risk factors assessed / the total number of risk factors assessments expected) × 100%. The total number of risk factors that should be assessed in accordance with the risk factors assessment sheet, including hearing impairment, visual impairment, pain, use of anesthetic or sedatives, mechanical ventilation, indwelling catheter, infection, immobility, sleep disorders, and no family members visit. The total adherence to risk factors assessment per day is calculated as (the number of risk factors assessed for all cared patients/ the total number of risk factors assessments expected for all cared patients) × 100%.

c.Adherence to ICU delirium assessment is defined as (the number of ICU delirium assessed / the total number of ICU delirium assessment expected) × 100%. The total number of ICU delirium assessments for each patient is calculated two times (8 to 10 A.M. and 4 to 6 P.M.) per day. The total adherence to ICU delirium assessment per day is calculated as (the number of ICU delirium assessed for all cared patients/ the total number of ICU delirium assessments expected for all cared patients) × 100%.

• Z.177: Is actual delirium risk factors assessment meant here?

Response: Yes, it is the actual number of delirium risk factors assessed. We revised the sentence more specifically to make it clear how to calculate the adherence to risk factors assessment. As following:

b.Adherence to risk factors assessment is defined as (the number of risk factors assessed / the total number of risk factors assessments expected) × 100%. The total number of risk factors that should be assessed in accordance with the risk factors assessment sheet, including hearing impairment, visual impairment, pain, use of anesthetic or sedatives, mechanical ventilation, indwelling catheter, infection, immobility, sleep disorders, and no family members visit. The total adherence to risk factors assessment per day is calculated as (the number of risk factors assessed for all cared patients/ the total number of risk factors assessments expected for all cared patients) × 100%.

• Z.172: Is this about adherence to the delirium risk factors assessment in general, or to the assessment of individual risk factors?

Response: In this revised manuscript, we described both adherence to the assessment of individual risk factors and adherence to the delirium risk factors assessment in general. In the finally statistical analysis, adherence to the delirium risk factors assessment in general is used to represent adherence to the delirium risk factors assessments.

• Z.225: You assume an adherence of 80% for the AI-AntiDelirium group and 50% for the paper-based PADIS group and refer to the study by Trogrlic et al. 2018. This assumption should be explained.

Response: A sample size calculation should specify a minimum treatment effect that is both clinically important and likely to be achievable based on evidence from previous trials of similar interventions. In the study by Trogrlic et al. (2018), they designed an e-learning program for delirium screening (Phase II) and delirium guideline (Phase III). The primary outcome was adherence changes to delirium guidelines recommendations, and the study showed improved health professionals’ adherence to delirium guidelines. The average adherence to delirium guidelines (delirium screening, sedation assessments, and light sedation; Top 3 adherence) after the e-learning program for guideline implementation was about 80%, and the average adherence to delirium guidelines for the baseline group was about 50%. Therefore, we assume an adherence of 80% for the AI-AntiDelirium group (intervention group) and 50% for the paper-based individual interventions (control group).

---

## [Decision Letter · Decision Letter 1]

25 Aug 2023

PONE-D-23-02468R1Impact of Artificial Intelligence Assisted Prevention and Management for Delirium on guideline adherence and cognitive load in Intensive  Care Unit nurses (AID-ICU): protocol of a multicenter, cluster randomized trialPLOS ONE

Dear Dr. Wu,

Thank you for submitting your manuscript to PLOS ONE. After careful consideration, we feel that it has merit but does not fully meet PLOS ONE’s publication criteria as it currently stands. Therefore, we invite you to submit a revised version of the manuscript that addresses the points raised during the review process.

We look forward to receiving your revised manuscript.

Kind regards,

Sascha Köpke

Academic Editor

PLOS ONE

Journal Requirements:

Reviewers' comments:

Reviewer's Responses to Questions

**Comments to the Author**

1. Does the manuscript provide a valid rationale for the proposed study, with clearly identified and justified research questions?

Reviewer #2: Partly

2. Is the protocol technically sound and planned in a manner that will lead to a meaningful outcome and allow testing the stated hypotheses?

Reviewer #2: Yes

3. Is the methodology feasible and described in sufficient detail to allow the work to be replicable?

Reviewer #2: No

4. Have the authors described where all data underlying the findings will be made available when the study is complete?

Reviewer #2: Yes

5. Is the manuscript presented in an intelligible fashion and written in standard English?

Reviewer #2: Yes

6. Review Comments to the Author

You may also provide optional suggestions and comments to authors that they might find helpful in planning their study.

Reviewer #2: Dear authors,

I am very pleased that you have implemented my suggestions. There are still some ambiguities that are probably easy to implement.

Z.72 - 75: The cited literature by Han et al. 2021 describes communication via text messaging in an intensive care unit .

Z.86 With regard to the development of the AI-AntiDelirium, you refer to another publication as well as to 5 steps of the development of the AI-AntiDelirium. Were the results of the development process published? If not, would a separate publication be interesting?

Is the AI-AntiDelirium really artificial intelligence? Is it not rather a Clinical Decision Support System (CDSS) that acts on the basis of predefined IF-Then rules? You might want to reconsider the term Artificial Intelligence in the title and in the acronym "AI-AntiDelirium" and in other places.

203: The calculation of adherence is now more comprehensible. However, the adherence cannot be between 0 and 1 if the calculation of the formulas a, b, c is done according to a different systematic.

257: The description of the level of knowledge of the nurses rather belongs to the chapter Outcomes.

243: You assume that the recruitment of nurses will be completed by the end of June 2023. What is the current status of recruitment? Are changes to the protocol still feasible?

Best regards

7. PLOS authors have the option to publish the peer review history of their article (what does this mean?). If published, this will include your full peer review and any attached files.

Reviewer #2: No

---

## [Author Response · Author response to Decision Letter 1]

29 Aug 2023

Reviewer #2: Dear authors,

I am very pleased that you have implemented my suggestions. There are still some ambiguities that are probably easy to implement.

Z.72 - 75: The cited literature by Han et al. 2021 describes communication via text messaging in an intensive care unit .

Response: We replaced the cited literature by Han et al. 2021 as Ceballos-Vasquez et al:

Ceballos-Vasquez P, Rolo-Gonzalez G, Hernandez-Fernaud E, Diaz-Cabrera D, Paravic-Klijn T, Burgos-Moreno M. Psychosocial factors and mental work load: a reality perceived by nurses in intensive care units. Rev Lat Am Enfermagem. 2015;23(2):315-22. doi: 10.1590/0104-1169.0044.2557.

Z.86 With regard to the development of the AI-AntiDelirium, you refer to another publication as well as to 5 steps of the development of the AI-AntiDelirium. Were the results of the development process published? If not, would a separate publication be interesting?

Response: We appreciate the reviewer very much for your positive comments and interest in the development of the AI-AntiDelirium. Detailed information about the Development of an Artificial Intelligence-Assisted Prevention and Management for Delirium (AI-AntiDelirium) is under review in a peer-reviewed journal.

Is the AI-AntiDelirium really artificial intelligence? Is it not rather a Clinical Decision Support System (CDSS) that acts on the basis of predefined IF-Then rules? You might want to reconsider the term Artificial Intelligence in the title and in the acronym "AI-AntiDelirium" and in other places.

Response: Thanks a lot for your very helpful comment. As you suggested, we modified the Title as following: 

Impact of Clinical Decision Support System Assisted Prevention and Management for Delirium on guideline adherence and cognitive load in Intensive Care Unit nurses (CDSSD-ICU): protocol of a multicenter, cluster randomized trial.

In addition, we revised the the acronym "AI-AntiDelirium" into "CDSS-AntiDelirium".

203: The calculation of adherence is now more comprehensible. However, the adherence cannot be between 0 and 1 if the calculation of the formulas a, b, c is done according to a different systematic.

Response: Thanks a lot for your kindly comment. We revised the calculation of adherence as following:

Complete adherence rate is recorded as 100%, and complete non-adherence rate is recorded as 0%. Adherence rate is between 0 and 100 according to the following formula.

257: The description of the level of knowledge of the nurses rather belongs to the chapter Outcomes.

Response: As you suggested, we moved the level of knowledge of the nurses into the Outcome measures section.

243: You assume that the recruitment of nurses will be completed by the end of June 2023. What is the current status of recruitment? Are changes to the protocol still feasible?

Response: The current status of recruitment is finished, between November 2022 and June 2023, 38 nurses (mean [SD] age, 33.29 [0.88] years; 33 [86.8%] female) are included in the intervention group, and 42 nurses (mean [SD] age, 33.52 [0.79] years; 39 [92.9%] female) are recruited in the control group. Changes to the protocol are still feasible, we continue to collect the outcomes among recruiting ICU nurses.

---

## [Editor Report · Decision Letter 2]

11 Sep 2023

PONE-D-23-02468R2Impact of Clinical Decision Support System Assisted Prevention and Management for Delirium on guideline adherence and cognitive load in Intensive  Care Unit nurses (CDSSD-ICU): protocol of a multicenter, cluster randomized trialPLOS ONE

Dear Dr. Wu,

Thank you for submitting your manuscript to PLOS ONE. After careful consideration, we feel that it has merit but does not fully meet PLOS ONE’s publication criteria as it currently stands. Therefore, we invite you to submit a revised version of the manuscript that addresses the points raised during the review process.

We look forward to receiving your revised manuscript.

Kind regards,

Sascha Köpke

Academic Editor

PLOS ONE

Journal Requirements:

Additional Editor Comments:

The paper has again been reviewed and the reviewer is mostly happy with the manuscript.

However, before publication,there are three more aspects to be addressed:

(1) Most importantly, please apply another language check preferably by a professional language editor as there are still a number of inadequate expressions and language use is often incorrect.

(2) In l.139, you state that “Investigators who will enroll participants are not made aware of the randomization list.” Please be clearer about this process. As “post-randomization recruitment bias” is a common risk of bias in cluster randomized studies, it is important to know, who is recruiting patients and determining their eligibility.

(3) Under “Patient and public involvement” (l.347ff.), please be more specific. Information is not involvement. So if there is no active role in the sense of PPI, please state this clearly.

---

## [Author Response · Author response to Decision Letter 2]

10 Oct 2023

Additional Editor Comments:

The paper has again been reviewed and the reviewer is mostly happy with the manuscript.

However, before publication,there are three more aspects to be addressed:

(1) Most importantly, please apply another language check preferably by a professional language editor as there are still a number of inadequate expressions and language use is often incorrect.

Response: Thanks a lot for your comment. As you suggested, we polished the manuscript by a professional language expert. 

(2) In l.139, you state that “Investigators who will enroll participants are not made aware of the randomization list.” Please be clearer about this process. As “post-randomization recruitment bias” is a common risk of bias in cluster randomized studies, it is important to know, who is recruiting patients and determining their eligibility.

Response: We provided more information about this process, as following:

Investigators who will enroll participants will not be made aware of the randomization list. Once the ICUs (cluster) that will take part in the study have been determined, randomization will be performed, considering the size of each ICU and the type of disease treated. The following criterion will be considered in this process: whether the center is classified as having a high incidence of delirium. Investigators will recruit participants following predefined inclusion and exclusion criteria. 

(3) Under “Patient and public involvement” (l.347ff.), please be more specific. Information is not involvement. So if there is no active role in the sense of PPI, please state this clearly.

Response: We provided more detailed information in the “Patient and public involvement”, as following:

Patients and/or the public (including clinical professionals) will be involved in the study design, or conduct, or reporting of this research. A participatory workshops carried out in October 2022 focused on collecting their opinions and solving problems in the design of the protocol.

---

## [Editor Report · Decision Letter 3]

24 Oct 2023

Impact of Clinical Decision Support System Assisted Prevention and Management for Delirium on guideline adherence and cognitive load among  Intensive  Care Unit nurses (CDSSD-ICU): protocol of a multicentre, cluster randomized trial

PONE-D-23-02468R3

Dear Dr. Wu,

We’re pleased to inform you that your manuscript has been judged scientifically suitable for publication and will be formally accepted for publication once it meets all outstanding technical requirements.

Kind regards,

Sascha Köpke

Academic Editor

PLOS ONE

Additional Editor Comments (optional):

The authors have adequately responded to the final recommendations.
---

## [Editor Report · Acceptance letter]

13 Nov 2023

PONE-D-23-02468R3 

Impact of Clinical Decision Support System Assisted Prevention and Management for Delirium on guideline adherence and cognitive load among Intensive Care Unit nurses (CDSSD-ICU): protocol of a multicentre, cluster randomized trial 

Dear Dr. Wu:

I'm pleased to inform you that your manuscript has been deemed suitable for publication in PLOS ONE. Congratulations! Your manuscript is now with our production department. 

Kind regards, 

on behalf of

Professor Sascha Köpke 

Academic Editor

PLOS ONE